# Statin prescription in patients with chronic obstructive pulmonary disease and risk of exacerbations: a retrospective cohort study in the Clinical Practice Research Datalink

Margaret C Smith ,[1,2] Helen Frances Ashdown ,[1] James Peter Sheppard ,[1] Christopher C Butler ,[1] Clare Bankhead [1,2]

[1]Nuffield Department of Primary Care Health Sciences, University of Oxford, Oxford, UK
[2]NIHR Oxford Biomedical Research Centre, Oxford University Hospitals NHS Foundation Trust, Oxford, UK

**Correspondence to**
Dr Margaret C Smith;
margaret.smith@phc.ox.ac.uk

## ABSTRACT

**Objective** Observational studies have suggested a beneficial effect of taking statins on frequency of chronic obstructive pulmonary disease (COPD) exacerbations. However, clinical trials of statins in people with COPD did not confirm those results. This study aimed to investigate this association using a methodological approach, which reduces the biases associated with some previous observational study designs.

**Design** Retrospective cohort study comparing new-users of statins with non-users.

**Setting** General practices in England contributing to the Clinical Practice Research Datalink in 2007–2017, with linkage to data on Hospital Episode Statistics inpatient episodes.

**Participants** 48 124 people with COPD, aged over 40 years, who had not been prescribed statin in the previous year.

**Exposure** Participants became new-users of statins at their first prescription for a statin during follow-up. They were then assumed to remain statin users. Statin users were compared with non-users.

**Outcomes** Primary outcomes were COPD exacerbation, or severe exacerbation requiring hospitalisation. Secondary outcomes were death from any cause (for comparison with other studies) and urinary tract infection (negative-control). Maximum follow-up was 3 years. Adjusted HR were calculated using time-dependent Cox regression. The Andersen-Gill model was used for recurrent exacerbations. Covariates included demographic variables, variables related to COPD severity, cardiovascular comorbidities as time-dependent variables, and other comorbidities at baseline.

**Results** 7266 participants became new-users of statins over an average 2.5 years of follow-up. In total, 30 961 people developed an exacerbation, 8110 severe exacerbation, 3650 urinary tract infection and 5355 died. Adjusted HR (95% CI) in statin users compared with non-users were first exacerbation 1.01 (0.96–1.06), severe exacerbation 0.92 (0.84–0.99), number of exacerbations 1.00 (0.97–1.04), urinary tract infection 1.10 (0.98–1.23) and death 0.63 (0.57–0.70).

### Strengths and limitations of this study

► We used a large real-world cohort of over 48 000 people with chronic obstructive pulmonary disease who were statin naïve at study entry.
► The exposure was defined as new statin prescription in follow-up and analysed as a time-dependent variable. New user designs are less susceptible to prevalent-user bias.
► We included secondary outcomes and various levels of confounding adjustment to investigate some potential sources of bias.
► This study identified statin users from their first prescription of statins. We cannot be certain that the medication was taken and adherence may have declined over follow-up.
► As this was an observational study, we cannot be certain that residual confounding or other biases did not affect our results

**Conclusions** In this study of health records from a Primary Care database, statin use in people with COPD was not associated with a lower risk of COPD exacerbation.

## INTRODUCTION

Chronic obstructive pulmonary disease (COPD) is characterised by progressive and irreversible airflow limitation, together with sudden flare-ups or exacerbations. Frequent exacerbations accelerate loss of lung function and are associated with poorer quality of life and poor survival.[1] COPD is often associated with the presence of other comorbidities, of which cardiovascular disease (CVD) is one of the most significant.[2] Therefore, many people with COPD are likely to be recommended medication for primary or secondary prevention of CVD, including with statins.[3] Increased levels of biomarkers reflecting systemic

inflammation are also part of the pathology of COPD. Levels of inflammatory biomarkers increase during the acute phase of an exacerbation, and in individuals with stable COPD, elevated levels of these biomarkers are associated with higher frequency of exacerbations.[4 5] Statins have been found to have anti-inflammatory effects, as well as reducing cholesterol.[6] Therefore, there is a theoretical possibility that statins may have a beneficial role in COPD management over and above reducing cardiovascular risk.

Several observational studies have found an association between statin use and fewer exacerbations in COPD patients.[7 8] However, randomised controlled trials have failed to confirm any such associations.[9] Several explanations for the conflict of trial results with those from observational studies have been proposed. First, immortal time bias arises from misspecification of exposures so that there is a span of cohort follow-up during which, because of exposure definition, the outcome under study cannot occur.[10] Second, prevalent user bias occurs when risk of an outcome changes over time in a population exposed to a drug treatment.[11] Finally, the population and treatment in the clinical trials were very precisely defined and unlikely to be typical of the general population.[12]

A study design in which all patients are statin naïve at study entry, and in which initiation of statins is included as a time-dependent variable can reduce the likelihood of some of these biases. This study aimed to use such a study design to examine the association between statin prescription and incidence of COPD exacerbations in a general population of patients attending routine primary care.

## METHODS
### Study design
The study is a retrospective cohort study comparing new-users of statins with non-users. It was conducted within the Clinical Practice Research Datalink (CPRD) GOLD database, which is a collection of anonymised primary care records from ~700 UK practices. These data have been shown to be representative of the general UK population in terms of age, sex and ethnicity.[13] We used data from the three-quarters of CPRD English practices with linkage to the English Hospital Episode Statistics (HES), and mortality data from the UK Office for National Statistics and patient-level social deprivation indices.

### Patient and public involvement
As this is a retrospective database study, there are limited opportunities for patient and public involvement in study design or conduct. During the development of the protocol, we discussed experience of exacerbations with three people who had COPD themselves or had experience of caring for people with COPD.

### Population and exposure
We included patients aged at least 40 years, who consulted between January 2007 and July 2017, and had a preceding read code for COPD in their medical records within research-quality data. Research quality data fulfils basic CPRD-defined quality criteria at the practice (recorded death rates are within a predefined range based on the UK average and that there is continuity in recording of deaths and other events) and patient level (included patients are permanently registered at their practice with a valid birth year and registration date).[14] To confirm the presence of COPD, a forced expiratory volume in one second/forced vital capacity ($FEV_1/FVC$) of <0.7 had to also be recorded.[15] To ensure that patients were actively consulting and that there were enough data to identify baseline values of covariates, the index date was defined as the latest of the consultation date and the date of accumulating 2 years of research quality data in the current practice. Patients in the selected cohort were also all statin naïve as they were only included in the cohort if there had been no statin prescription in the 12 months before the index date. They were defined as newly prescribed users of statins once a prescription was issued on the index date or during follow-up.

### Outcomes
Patients were followed-up until they were censored after 3 years, or until the end of the study period or they died or left their practice or their practice stopped contributing data to CPRD GOLD, if any of these occurred within 3 years. Follow-up was limited to 3 years because we expected adherence to statins to drop-off over time.[16]

Our primary outcomes were any COPD exacerbation and severe exacerbation requiring hospitalisation. An acute exacerbation of COPD in the Primary Care record was defined as a code for acute exacerbation, code for lower respiratory tract infection, prescriptions for 5–14 days of exacerbation-specific antibiotics and oral steroid on the same date, or a symptom of exacerbation together with a prescription for exacerbation-specific antibiotics or oral steroids.[17] Exacerbations were also identified from the HES inpatient record, from a COPD or acute respiratory codes as the primary diagnosis of hospitalisation (ICD-10 J00, J06, J09-18, J20-22, J40-44 and J96), or from a code for acute exacerbation of COPD anywhere in the hospitalisation record (J44.0 and J44.1).[18] We identified all exacerbations in follow-up. To prevent double counting, events indicating exacerbations within 14 days of each other were assumed to describe the same exacerbation, with the exacerbation date taken from the first code. A severe exacerbation was any exacerbation with a relevant code, as described above, in the HES inpatient data at some point in its duration.

Secondary outcomes were death from any cause, and urinary tract infection (UTI). Death was chosen so that we could compare the observed association with results from randomised controlled trials in the general population. UTI recorded in Primary Care data was included as a

negative-control outcome because we did not expect any association with statin use. A strong association between statin use and UTI might indicate some bias in the study design.

## Covariates

We assessed potential confounders at the index date and time dependently. For the purpose of adjustment in regression analyses, covariates were combined into groups: lifestyle and demographic factors, markers of COPD severity, CVD-related variables and other co-morbidities or health-related variables.

► Lifestyle and demographic variables (basic adjustment) included gender, age at index, deprivation (as quintile of Index of Multiple Deprivation (IMD)) and smoking status before index.

► Covariates related to COPD severity were $FEV_1$ percent of predicted, number of COPD exacerbations, severe exacerbation requiring hospitalisation, regular prescription of COPD-related medication (short-acting bronchodilator, long-acting bronchodilator, inhaled corticosteroid, nebuliser) and receiving oxygen therapy.[19 20] The most recent $FEV_1$ percent of predicted in the 2 years prior to index was extracted. All other variables were assessed in the 1 year before index. COPD exacerbations were defined as for outcomes. Regular prescription was defined as at least three prescriptions in the year.

► CVD-related variables were previous CVD, diabetes and hypertension. These variables were assessed as if they had occurred ever before index and time-dependently during follow-up. We included this group of variables as time-dependent covariates in main analyses because they are very likely to be associated with statin exposure. CVD was defined as a diagnosis code for one of coronary heart disease, stroke, heart failure or peripheral artery disease in the Primary Care data or in the linked HES inpatient data. Hypertension was defined as a diagnosis code or a prescription for an antihypertensive medication. We also included body mass index (BMI) (the most recent measurement in 2 years before index) in this group of covariates.

► Comorbidities and other health-related variables including asthma, other respiratory disease (bronchiectasis, cystic fibrosis, alpha-1 antitrypsin-deficiency and interstitial lung disease), lung cancer, rheumatoid arthritis, gastro-oesophageal reflux disease and osteoporosis were assessed from codes in Primary Care and HES inpatient data recorded at any time before the index date. Depression, influenza vaccination and number of doctor or nurse consultations were assessed in Primary Care data in the year before index.

## Statistical analysis

We used Cox regression to investigate the association of statin use with time to first event for exacerbation, severe exacerbation, death or UTI. The proportional hazards assumption was assessed for the time-independent covariates using graphical methods (-ln –ln of survival curves). For analyses of recurrent exacerbation, we used the Andersen and Gill model.[21] Statin use was defined as a binary time-dependent variable. All patients were non-users of statin immediately before the index date. They became new-users of statins at their first prescription of statin on the index date or during follow-up. In main analyses, we took an intention to treat approach in which people were assumed to remain statin users from the initial prescription until they were censored after a maximum of 3 years of follow-up. We censored after 3 years because we expected adherence to statins to fall over time, which could dilute the observed effect size. The analyses were repeated with combinations of the different sets of covariates to try to understand the role of confounding. All analyses were conducted in Stata V.16 (StataCorp, Texas, USA).

Age at index and BMI were included in analyses as continuous covariates. $FEV_1$ percent of predicted was included as a categorical variable (≥80%, 50≤80%, 30≤50%, <30%). Quintile of IMD, number of exacerbations (0, 1, 2, 3, 4, ≥5) and smoking status (never, ex, current) were also included as categorical variables. Other covariates were included as binary variables. Time-dependent variables were incorporated by splitting patient records at the date of the first prescription for statin or when CVD, diabetes or hypertension were first recorded.

Missing values of BMI or $FEV_1$ percent of predicted were imputed with multiple imputation by chained equations, using a linear regression model for imputing $FEV_1$ percent of predicted and predictive mean matching for BMI, and including all variables to be included in planned analyses (including outcomes). Thirty imputed data sets were calculated and estimates from regressions were pooled using Rubin's rules. People with missing smoking data were assumed non-smokers. Missing values of other variables were interpreted as a null value, for example, no prescription or no comorbidity. We also repeated analyses restricted to people with complete data. We did not find any substantive difference in results (data not shown).

In secondary analyses, we censored 90-days after the last recorded statin prescription to investigate the effect of non-adherence on associations. In main analyses that were adjusted for CVD-related variables, we adjusted for these variables at index and time-dependently. A post hoc sensitivity analysis was also conducted for the death outcome in which we adjusted for CVD variables assessed at baseline rather than the time-dependent CVD variables.

## RESULTS
### Baseline characteristics

A total of 8 375 283 people were available with complete linkage within the CPRD GOLD database (figure 1). Within these, we identified 76 702 people aged 40+ years who fulfilled the definition of a COPD diagnosis, had a recent record of consulting their GP and had data in

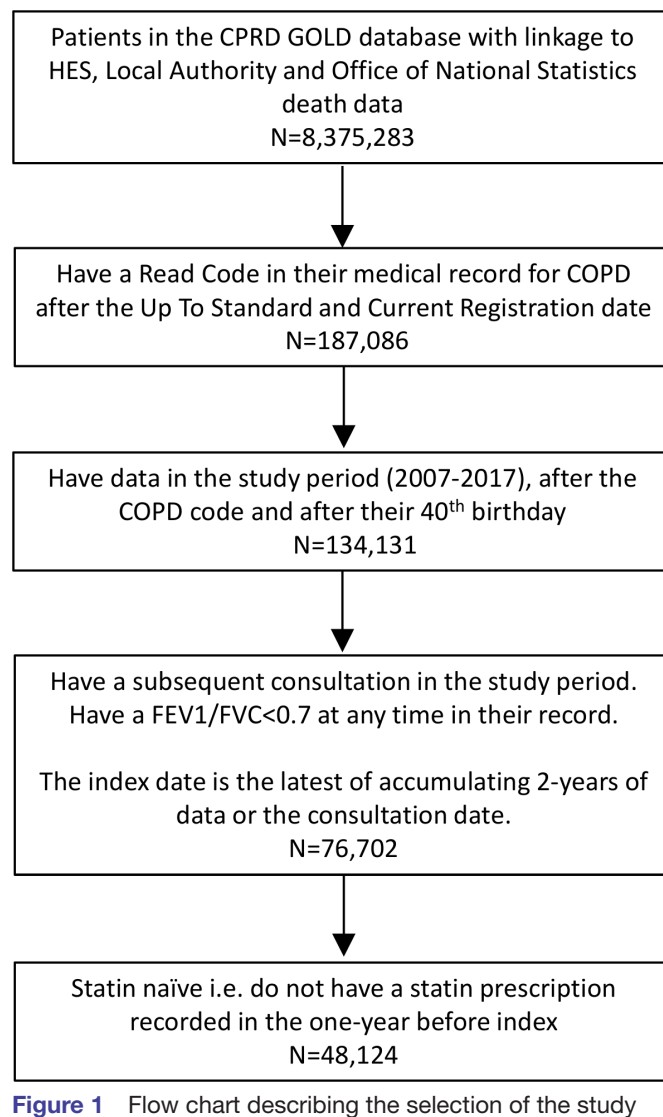

**Figure 1** Flow chart describing the selection of the study population from the Clinical Practice Research Datalink (CPRD) GOLD database. $FEV_1$/FVC is the ratio of forced expiratory volume in one second to forced vital capacity.

the study period. Of these, 48 124 were statin naïve and were included in the cohort. Of these, 7266 had at least one statin prescription on the index date or subsequently during a maximum of 3 years of follow-up. For 40 858, there was no record of a statin prescription in follow-up.

People who subsequently started on statins, that is, became new-users of statins, had different baseline characteristics compared with those who did not (table 1). They were less likely to be female (42% compared with 50%). They were also more likely to have social deprivation above the median (57% vs 54%) but slightly less likely to be never smokers (8% vs 10%). Variables related to COPD severity indicated that COPD was on average slightly less severe at baseline in the new statin users. Those who became statin users had a slightly higher BMI at baseline (27 vs 26 kg m$^{-2}$) and considerably higher prevalence of hypertension (50% vs 39%), CVD (14% vs 8%) and diabetes (6% vs 2%) at baseline. More had hypertension, CVD or diabetes newly recorded in follow-up.

**Table 1** Descriptive statistics for the study population according to whether they received a statin prescription in follow-up, that is, became new-users of statins

| Variable | Prescription for statin in follow-up | |
| --- | --- | --- |
| | No (n=40 858) | Yes (n=7266) |
| Index date (median) | March 2008 | July 2007 |
| Lifestyle and demographic | | |
| Female | 20 375 (50%) | 3045 (42%) |
| Mean (SD) age (years) | 67(12) | 68(10) |
| Deprivation above median (IMD 2007) | 22 109 (54%) | 4174 (57%) |
| Smoking status | | |
| Never | 4153 (10%) | 604 (8%) |
| Ex | 19 978 (49%) | 3661 (50%) |
| Current | 16 524 (40%) | 2973 (41%) |
| Missing | 203 (0%) | 28 (0%) |
| Related to COPD severity | | |
| Mean (SD) $FEV_1$ (percent of predicted) | 63(21) | 65(21) |
| Missing | 11 278 (28%) | 2062 (28%) |
| $FEV_1$ percent of predicted | | |
| ≥80 | 6073 (15%) | 1123 (15%) |
| ≥50≤80 | 15 152 (37%) | 2788 (38%) |
| ≥30≤50 | 6966 (17%) | 1100 (15%) |
| <30 | 1389 (3%) | 193 (3%) |
| Missing | 11 278 (28%) | 2062 (28%) |
| Number of COPD exacerbations in 1 year before index | | |
| 0 | 20 336 (50%) | 3785 (52%) |
| 1 | 11 154 (27%) | 1914 (26%) |
| 2 | 5073 (12%) | 884 (12%) |
| 3 | 2237 (5%) | 346 (5%) |
| 4 | 1024 (3%) | 175 (2%) |
| ≥5 | 1034 (3%) | 162 (2%) |
| At least one severe exacerbation in 1 year before index | 2981 (7%) | 499 (7%) |
| Regular prescription (3+) of COPD medication in 1 year before index | | |
| Short acting bronchodilator | 20 936 (51%) | 3661 (50%) |
| Long acting bronchodilator | 18 068 (44%) | 3075 (42%) |
| Inhaled corticosteroid | 19 136 (47%) | 3323 (46%) |
| Nebuliser | 1654 (4%) | 230 (3%) |
| On oxygen therapy | 423 (1%) | 49 (1%) |
| Related to cardiovascular disease | | |
| Diabetes | | |
| Recorded before index date | 1007 (2%) | 462 (6%) |

Continued

**Table 1** Continued

| Variable | Prescription for statin in follow-up | |
| --- | --- | --- |
| | No (n=40 858) | Yes (n=7266) |
| New over follow-up | 482 (1%) | 693 (10%) |
| Hypertension | | |
| Recorded before index date | 16 093 (39%) | 3661 (50%) |
| New over follow-up | 4632 (11%) | 2066 (28%) |
| Cardiovascular disease | | |
| Recorded before index date | 3241 (8%) | 991 (14%) |
| New over follow-up | 1812 (4%) | 1984 (27%) |
| Body mass index (kg m$^{-2}$) mean (SD) | 26(6) | 27(6) |
| Missing | 10 838 (27%) | 1679 (23%) |
| Comorbidities and other health-related recorded before index date | | |
| Respiratory comorbidities | | |
| Asthma | 10 818 (26%) | 1890 (26%) |
| Other respiratory disease | 1289 (3%) | 195 (3%) |
| Other comorbidities | | |
| Lung cancer | 331 (1%) | 46 (1%) |
| Depression in 1 year before index | 2808 (7%) | 474 (7%) |
| Rheumatoid arthritis | 700 (2%) | 123 (2%) |
| Gastro-oesophageal reflux disease | 4966 (12%) | 995 (14%) |
| Osteoporosis | 2113 (5%) | 300 (4%) |
| Influenza vaccination in 1 year before index | 27 143 (66%) | 5211 (72%) |
| Regular (5+) contacts with practice staff in 1 year before index | | |
| GP | 22 525 (55%) | 4153 (57%) |
| Nurse | 8979 (22%) | 1772 (24%) |

Baseline characteristics are given for all variables. New cases over follow-up are also given for cardiovascular disease and related conditions. Statistics are n (%) unless otherwise stated.
COPD, chronic obstructive pulmonary disease; FEV$_1$, forced expiratory volume in one second percent of predicted; GP, general practitioner; IMD, index of multiple deprivation.

New statin-users were also more like to have an influenza vaccination in the year before baseline (72% vs 66%), and more likely to make 5+ visits to a GP (57% vs 55%) or nurse (24% vs 22%).

## Associations of statin use with outcomes
The mean follow-up was 2.5 years; 17 350 (36%) people were censored before the end of the maximum 3 years of follow-up, due to death (5355 (11%)), end of the study period (1973 (4%)), or leaving their current GP practice or the end of downloaded data in CPRD from their practice (10 022 (21%)). Over follow-up, there were 106 961 exacerbations recorded; 30 961 (64%) people had at least one exacerbation, 8110 (17%) were hospitalised for a severe exacerbation and 3650 (8%) had at least one UTI diagnosed. The mean follow-up in nonusers of statins (to censoring, death or starting on statins) was 2.2 years (total 107 039 years). The mean follow-up after starting to use statins was 1.6 years (total 11 307 person years).

The first statin prescription was simvastatin in 6089 (84%) of the new users. It was atorvastatin for 989 (14%), pravastatin for 133 (2%), rosuvastatin for 53 (1%) and fluvastatin for two people. The number of statin prescriptions per person was 9 (4–18) (median and IQR). Among those whose first prescription was for simvastatin, 397 (7%) had a prescription for a daily dose of 10 mg, 1822 (30%) for 20 mg, 3864 (64%) for 40 mg and 6 (<1%) for 80 mg.

### COPD exacerbation
There was no association between statins and COPD exacerbation after adjustment for basic demographic variables (HR 0.99, 95% CI 0.94 to 1.03) (table 2). Additionally adjusting for cardiovascular risk factors resulted in a small statistically significant beneficial effect of statins (HR 0.94, 95% CI 0.90 to 0.99) (table 2). The association was constant after further adjustment for variables related to severity of COPD (HR 1.03, 95% CI 0.99 to 1.08). This remained the case after adjustment for all covariates (HR 1.01, 95% CI 0.96 to 1.06) and in analyses censoring 90-days after the last statin prescription (table 3). A similar pattern of HR with different adjustments was seen for the recurrent COPD exacerbation outcome and for severe COPD exacerbations. The HR for recurrent COPD exacerbations adjusted for all covariates was 1.00 (95% CI 0.97 to 1.04). The HR for severe exacerbations was marginally significant after adjustment for all covariates (HR 0.92, 95% CI 0.84 to 0.99).

### Secondary outcomes
Taking statins was associated with a lower risk of death from any cause. The HR after basic adjustment was 0.85 (95% CI 0.77 to 0.93) (table 2). The association was considerably strengthened after adjusting for baseline and time-dependent CVD risk factors. After adjustment for all covariates, the HR was 0.63 (95% CI 0.57 to 0.70). In the post hoc sensitivity analysis in which adjustment was for all covariates but with CVD restricted to baseline values, the HR was 0.83 (95% CI 0.76 to 0.92). There was an increased risk of UTI in people taking statins after basic adjustment (HR 1.28, 95% CI 1.14 to 1.43). This was reduced and no longer statistically significant after adjusting for all covariates (HR 1.10, 95% CI 0.98 to 1.23).

## DISCUSSION
In this study of people with COPD who were followed up for a maximum of 3 years, we did not find evidence to

**Table 2** Hazard ratios (HR) for association of chronic obstructive pulmonary disease (COPD) exacerbation, urinary tract infection or death with statin use

| Outcome | Covariates in model* | HR | 95% CI | P value |
|---|---|---|---|---|
| Primary outcomes<br>COPD exacerbation (N events in statin users, non-users=1893, 29 068) | Unadjusted | 0.99 | 0.94 to 1.04 | 0.652 |
| | Basic | 0.99 | 0.94 to 1.03 | 0.583 |
| | Basic +CVD | 0.94 | 0.90 to 0.99 | 0.014 |
| | Basic +COPD | 1.03 | 0.99 to 1.08 | 0.168 |
| | Basic +COPD+CVD | 1.01 | 0.96 to 1.06 | 0.736 |
| | All covariates | 1.01 | 0.96 to 1.06 | 0.657 |
| COPD exacerbation, recurrent events (N events in statin users, non-users==10 352, 96 609) | Unadjusted | 1.02 | 0.98 to 1.06 | 0.359 |
| | Basic | 1.02 | 0.98 to 1.06 | 0.334 |
| | Basic +CVD | 0.94 | 0.91 to 0.98 | 0.002 |
| | Basic +COPD | 1.04 | 1.01 to 1.08 | 0.010 |
| | Basic +COPD+CVD | 1.00 | 0.97 to 1.03 | 0.999 |
| | All covariates | 1.00 | 0.97 to 1.04 | 0.885 |
| Severe COPD exacerbation (N events in statin users, non-users=722, 7388) | Unadjusted | 1.05 | 0.97 to 1.14 | 0.217 |
| | Basic | 1.00 | 0.92 to 1.08 | 0.977 |
| | Basic +CVD | 0.80 | 0.74 to 0.87 | <0.001 |
| | Basic +COPD | 1.07 | 0.99 to 1.16 | 0.087 |
| | Basic +COPD+CVD | 0.90 | 0.83 to 0.98 | 0.016 |
| | All covariates | 0.92 | 0.84 to 0.99 | 0.034 |
| Secondary outcomes<br>Death from any cause (N events in statin users, non-users=466, 4889) | Unadjusted | 0.87 | 0.79 to 0.96 | 0.005 |
| | Basic | 0.85 | 0.77 to 0.93 | 0.001 |
| | Basic +CVD | 0.58 | 0.53 to 0.64 | <0.001 |
| | Basic +COPD | 0.88 | 0.80 to 0.97 | 0.009 |
| | Basic +COPD+CVD | 0.63 | 0.57 to 0.69 | <0.001 |
| | All covariates | 0.63 | 0.57 to 0.70 | <0.001 |
| | Basic +COPD+baseline CVD | 0.83 | 0.76 to 0.92 | <0.001 |
| Urinary tract infection (N events in statin users, non-users=360, 3290) | Unadjusted | 1.19 | 1.07 to 1.33 | 0.002 |
| | Basic | 1.28 | 1.14 to 1.43 | <0.001 |
| | Basic +CVD | 1.07 | 0.95 to 1.20 | 0.273 |
| | Basic +COPD | 1.29 | 1.16 to 1.44 | <0.001 |
| | Basic +COPD+CVD | 1.09 | 0.97 to 1.22 | 0.159 |
| | All covariates | 1.10 | 0.98 to 1.23 | 0.112 |

*Covariates included in the models. Basic adjustment, lifestyle and demographic variables including gender, age, IMD quintile (categorical), smoking status (never, ex, current); COPD, severity of chronic obstructive disease (COPD) at the index date including exacerbation frequency (categorised as 0, 1, 2, 3, 4, 5+), severe exacerbation requiring hospitalisation, forced expiratory volume in one-second percent of predicted (categorised as 80, 50–<80, 30–<50,<30), regular prescriptions for COPD-related medications, received oxygen; CVD, cardiovascular disease (CVD) related variables at index and as time-dependent variables including diabetes, hypertension, CVD diagnosis, body mass index (at index); all covariates, included basic +COPD+CVD groups of covariates and also other comorbidities and health-related covariates recorded at index (see list in Methods). Baseline CVD, CVD-related variables at the index date (post hoc sensitivity analysis conducted for the death outcome only).

support a beneficial effect of statin use on incidence of COPD exacerbations. The HR were close to one whether we considered the first exacerbation or multiple exacerbations after study entry. Using statins was associated with a slightly lower (8%, 95% CI 1% to 16%) risk of severe

exacerbation requiring hospitalisation after adjusting for all covariates.

Our findings for the primary outcome of COPD exacerbation are consistent with those from most randomised controlled trials,[9 22 23] but contrast with those of previous

**Table 3** Hazard ratios (HR) for association of chronic obstructive pulmonary disease (COPD) exacerbation, death or urinary tract infection with statin use. Adherent statin users only.

| Outcome | N events | HR | 95% CI | P value |
|---|---|---|---|---|
| Primary outcomes | | | | |
| COPD exacerbation | 17 633 | 1.02 | 0.94 to 1.09 | 0.688 |
| COPD exacerbation, recurrent events | 102 342 | 1.00 | 0.96 to 1.04 | 0.995 |
| Secondary outcomes | | | | |
| Severe COPD exacerbation | 4365 | 0.89 | 0.78 to 1.01 | 0.080 |
| Death | 2828 | 0.50 | 0.42 to 0.60 | <0.001 |
| Urinary tract infection | 2054 | 0.99 | 0.82 to 1.20 | 0.943 |

Follow-up was censored 90-days after the last statin prescription. HR are adjusted for all covariates.*
*Covariates included in the models. Variables included in basic adjustment; variables related to severity of chronic obstructive disease (COPD) at the index date; cardiovascular disease-related variables at index and as time-dependent variables; other comorbidities and health-related covariates recorded at index. See Methods and legend to table 2 for details.

observational studies which have found 30%–40% lower risk of exacerbation with taking statins.[7][8] STATCOPE, the larger trial (885 participants) of simvastatin versus placebo for prevention of exacerbations among patients with moderate–severe COPD but no other indications for statin use was stopped early due to lack of benefit of statins on exacerbation rate.[23] The much smaller RODEO trial of rosuvastatin in patients with verified COPD but not CVD also failed to find any benefit of statin on exacerbation rate.[22] Recently published results from another small trial do however indicate some benefit of statin treatment on COPD exacerbation and mortality.[24] An explanation put forward by the authors for these results is that the study population was nearer 'real world' than in previous trials as it included patients with subclinical cardiovascular disease. CVD is strongly linked to COPD,[25][26] and so some hospital admissions for severe exacerbation may have been attributable to CVD. The beneficial effect of statins may has arisen as a knock-on effect on severity of COPD exacerbations via reduced incidence of CVD. Observational studies and randomised controlled trials have found possible beneficial effects of statin use on development of many other diseases but as with statins and COPD exacerbations, most of these findings need further rigorous investigation.[27][28]

We included UTI as a negative-control outcome. In other words, we would not expect an association with statin use if confounding was controlled for and if the study was free from other biases. We found that the rate of UTI was slightly increased in statin users, but this association became non-significant after adjustment for cardiovascular risk factors.

Taking statins was associated with a 15% (95% CI 7% to 23%) lower risk of death after basic adjustment for demographic variables and a 37% lower risk of death (95% CI 30% to 43%) after adjusting for all confounders. Our observed effect size is somewhat larger than that has been estimated in trials conducted in the general population, which is about 10%.[27][29] Healthy user biases, or the reverse in which COPD patients not on statins have poorly treated comorbid CVD which contributes to worse outcomes may have contributed to our observation.[12] Very ill patients with a higher risk of death or COPD exacerbation may not have been prescribed statins: patients in our study population who were prescribed statins had on average slightly milder COPD, and patients with end-stage COPD being managed palliatively are unlikely to be initiating a statin.

Those initiating statins had much higher prevalence of CVD-related variables at index date and incident over follow-up so we adjusted for time-dependent CVD. However, we may have inadvertently biased some associations by including new CVD diagnoses after statin initiation, because CVD risk is also affected by statin exposure.[30] Therefore, time-dependent CVD might also be a mediator on the path between statin use and outcomes. This raises the possibility that all HR that were adjusted for time-dependent CVD variables may have been subject to some degree of over-adjustment. The post hoc sensitivity analysis of the death outcome adjusting for baseline CVD but not time-dependent CVD resulted in an estimate of 17% (95% CI 8% to 24%) lower risk of death, which is closer to expectations. However, this estimate is also likely to be biased because it does not allow for confounding by CVD developed after the index date.

### Strengths and limitations of this study
Strengths of our study are the use of a large statin-naïve cohort with statin exposure included as a time-dependent exposure to minimise immortal time bias and prevalent user bias; adjustment for a wide range of potential confounding variables and inclusion of control outcomes to investigate residual confounding.

The CPRD GOLD database is based on Primary Care data and therefore our study population is more generalisable to real-world clinical practice than selected populations recruited for clinical trials. CPRD GOLD has been used previously for a wide range of studies, and has been shown to be representative of the UK population and

many variables extracted from the data have been validated externally including those related to COPD.[15 17 18]

Studies using databases of electronic health records also have their limitations. Non-adherence to prescribed statin medications or use of over-the-counter statins could potentially have reduced the size of any observed effect of statins on outcomes. Our assessment of statin use was based on statin prescriptions made in Primary Care. We cannot be certain that medications were dispensed or that patients were taking their medication. One study of individuals with COPD found that the proportion of prescriptions for COPD maintenance medications that were picked up was quite low, as was adherence to these medications.[31] Furthermore, low-dose statins have been available over-the-counter in the UK since 2004, and these would not be recorded in Primary Care records. Over-the-counter, statin medication is limited to 10 mg simvastatin but most Primary Care prescriptions for statins in our study were for 40 mg simvastatin. In addition, the support for over-the-counter statins by pharmacists and clinicians has been limited, and the take-up has been very low.[32 33]

COPD exacerbations are recorded in a number of different ways in Primary Care records and are therefore not straightforward to identify. We used a validated algorithm to identify exacerbations, but there is likely to have been some misclassification.[17] Non-differential misclassification is expected to move effect sizes towards the null. We also learnt from discussions with patient and public involvement volunteers that some exacerbations might be treated in accident and emergency departments. However, even though these data are available, it may be difficult to identify exacerbations in them, and no methodology has been validated yet, so by not including accident and emergency data, we may have missed COPD exacerbations, although documentation should be sent to the patient's GP. Missing data are potentially an issue in electronic health record databases: in particular, we did not have an estimate of $FEV_1$ percent predicted prior to index for 28% of included patients.

Other limitations of this study relate to the use of an observational study design. People starting on statins are likely to be different from those not doing so. We included a wide range of confounding variables in our models, nevertheless, we cannot be certain that there is not still some residual confounding. As discussed above, we included variables related to CVD as time-dependent covariates, which may have led to some degree of overadjustment. We included secondary outcomes to help to identify potential sources of bias, but we cannot be certain that our results are not affected by other sources of bias that we have not considered.

## CONCLUSION

We did not find evidence to support an association between statin prescription and reduced risk of exacerbation in people with COPD. These data suggest that in people with COPD and no other indication, statins should not be prescribed in routine primary care to reduce COPD exacerbations. Our findings are consistent with the well-established benefit of statins for reducing death in patients with CVD risk factors, and confirm this in a COPD population.

**Acknowledgements** We thank the Patient and Public Involvement volunteers for their time and for sharing their knowledge.

**Contributors** MCS planned and conducted the study, carried out statistical analyses and drafted the manuscript. HFA provided clinical expertise on chronic obstructive pulmonary disease. MCS, HFA, JPS, CCB and CB had full access to all the data in the study, contributed to the study design, interpretation of results, revisions of the manuscript and approved the final manuscript. MCS is study guarantor.

**Funding** This study/project is funded by the National Institute for Health Research (NIHR) School for Primary Care Research (project reference 365). JS receives funding from the Wellcome Trust/Royal Society via a Sir Henry Dale Fellowship (ref: 211182/Z/18/Z). HA has been funded by an NIHR Doctoral Research Fellowship (DRF-2014-07-052). CB and MS are supported by the National Institute for Health Research (NIHR) Oxford Biomedical Research Centre (BRC) and by the NIHR Applied Research Collaborative Oxford and Thames Valley. JS is supported by an NIHR Oxford BRC Senior Fellowship.

**Competing interests** CB reports grants from the NIHR and grants from the NIHR Health Protection Research Unit during the conduct of the study; personal fees from Pfizer outside of the submitted work. JS reports grants from Wellcome Trust/Royal Society and from NIHR Oxford Biomedical Research Centre. HA has contributed to a COPD prescribing study using the CPRD and her academic department has received fees from Boehringer-Ingelheim for this work. CB reports grants from NIHR Oxford Biomedical Research Centre, grants from NIHR Applied Research Collaborative Oxford and Thames Valley during the conduct of the study.

**Patient and public involvement** Patients and/or the public were involved in the design, or conduct, or reporting or dissemination plans of this research. Refer to the Methods section for further details.

**Patient consent for publication** Not required.

**Ethics approval** CPRD has ethics approval from the Health Research Authority to support research using anonymised patient data. GP practices choose to share patient data (primary care data) with CPRD for public health research purposes. However, individual patients can opt out of sharing their data with CPRD. The project protocol was approved by the CPRD Independent Scientific Advisory Committee (protocol number 18_059A).

**Provenance and peer review** Not commissioned; externally peer reviewed.

**Data availability statement** Data may be obtained from a third party and are not publicly available. This study is based on data from the CPRD obtained under licence from the UK Medicines and Healthcare products Regulatory Agency. Information on how to access the data used for these analyses is available at https://www.cprd.com/home/. The work uses data provided by patients and collected by the NHS as part of their care and support and would not have been possible without access to this data. The NIHR recognises and values the role of patient data, securely accessed and stored, both in underpinning and leading to improvements in research and care.

**ORCID iDs**
Margaret C Smith http://orcid.org/0000-0002-0946-742X
Helen Frances Ashdown http://orcid.org/0000-0002-7758-7095
James Peter Sheppard http://orcid.org/0000-0002-4461-8756
Christopher C Butler http://orcid.org/0000-0002-0102-3453
Clare Bankhead http://orcid.org/0000-0003-1588-3849

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
