## [Reviewer comments · BMJ Open]

ARTICLE DETAILS

TITLE (PROVISIONAL)	Statin prescription in patients with chronic obstructive pulmonary disease and risk of exacerbations: a retrospective cohort study in the Clinical Practice Research Datalink
AUTHORS	Smith, Margaret; Ashdown, Helen; Sheppard, James; Butler, Christopher C.; Bankhead, Clare

VERSION 1 – REVIEW

REVIEWER	Lahousse, Lies Ghent University
REVIEW RETURNED	27-Apr-2021

GENERAL COMMENTS	This large, impressive study investigated the association between statins and COPD exacerbations in a general population of patients attending routine primary care. Major comments: -In the methods, please specify the definition of "research quality data in the current practice" and "exacerbation-specific antibiotics".-Was the combination of antibiotics and oral steroids on the same date required to meet the definition of exacerbations? I understand that antibiotics alone may complicate the negative control of UTI but it could have underestimated true exacerbation rate? Since in elderly COPD, deterioration due to UTI or due to exacerbation may be hard to distinguish, it is very interesting that UTI was taken as a negative control. However, I wondered how events indicating exacerbations AND UTI within 14 days of each other were classified? Moreover, were UTI and exacerbations analyzed as competing risks in time-dependent analyses? Interestingly, adding CVD to the models has an opposite effect for UTI compared to exacerbations. Why was FEV1 included as confounder in the association of statins and exacerbations? Was this an attempt to correct for COPD severity which was on average slightly less severe in new statin users? Chronic low-grade system inflammation in COPD is not well associated with FEV1. This could have influenced results if not completely controlled for. -How were exacerbation and UTI results when restricted to COPD patients with indications for statin treatment or previous CVD?-Follow-up was limited to 3 years because adherence to statins was indeed expected to drop-off over time. However, previous observational studies are suggestive towards beneficial (pleiotropic) effects in COPD after long-term usage? I assume that most statins were initiated between the index date and end of follow-up (confirmed by 1.6y mean time on statins), so the period of actual statin intake may have been too short? How was the time between index date and statin initiation allocated?
--

	-Moreover regarding adherence, since statin-users are more likely to be healthy / more adherent users (also reflected by their higher influenza vaccinations and visiting), they are also likely to be more adherent to COPD maintenance therapy and therefore be less exacerbation-prone (confirmed by having 2% less exacerbations the year before index)? Minor comments:  - More likely to be current smokers (8% vs. 10%)? - Please specify in results text which factors are included in the "variables related to severity of COPD" - Lowering mortality and severe exacerbation requiring hospitalisation are very important outcomes in COPD, which seem currently underestimated in the overall conclusions.
--	---

REVIEWER	Roudi, Raheleh University of Minnesota Twin Cities
REVIEW RETURNED	28-Apr-2021

GENERAL COMMENTS	In the current study, Smith et al evaluated the statin prescription in patients with chronic obstructive pulmonary disease and risk of exacerbations. The authors need to consider some potential points to improve the current version;  1) COPD could be a driving factor in lung cancer and the authors should add this point as an important aspect of this study due to unefficient treatment options in lung cancer using the related articles such as Jianwei Zhu, et al - 2017 - Alfredo Tartarone, et al - 2019 - Monireh Mohsenzadegan, et al - 2020 and Fausto Petrelli, et al - 2021. 2) The association of statins with other diseases such as lung cancer have been suggested. The authors need to add these related data such as Feng Li, et al - 2018 and so on.
---

REVIEWER	Hopkins, Raewyn Auckland University of Technology
REVIEW RETURNED	21-May-2021

GENERAL COMMENTS	General Comment: The researcher's present a retrospective cohort study comparing newly prescribed statin users, with non-statin users, following the selection of an index date for study start. The researchers used a selection of General Practices in England who contribute data to the Clinical Practice Research Datalink, which is linked to a hospital inpatient statistics. The observation period was only an average of 2.5 years. Specific comments:  1. Included in the cohort are patients with Asthma, other respiratory disease and lung cancer, all of which effect lung function results and the development of chest infections / exacerbations. These patients should be removed from the analysis. 2. Given that smoking is a leading cause of COPD and cardiovascular disease, patients with missing smoking history should be removed as to assume them to be non-smokers maybe be incorrect.
--

	3. In the introduction (paragraph 1, page 5), the authors mention inflammatory biomarkers in relation to COPD. It is not clear to this reviewer whether the authors looked inflammatory markers such as CRP levels before and after statin treatment. The authors should do this and discuss the results. 4. There are studies showing evidence that concurrent smoking may delay the anti-inflammatory effect of statins, especially in the airways. The authors should stratify the groups by current versus ex-smokers and report and discuss the results. 5. Table 1 (page 9-11), shows the exacerbations within the year before the index date. The authors should stratify the cohorts according to exacerbations the year before the index date and see if the lack of a statin association is in those with 1 or more exacerbation. 6. In a cohort study such as this, the authors should add in and analyse all exacerbations. Time to 1st exacerbations is not useful unless it is an RCT. The authors should explain and discuss why they dropped out moderate exacerbations from analysis. 7. The authors should discuss how a diagnosis of pneumonia was handled, that is was it considered to be an exacerbation in this study. 8. Given the cardiac profile of the cohorts, the authors should discuss how a diagnosis of heart failure that received an antibiotic was handled, e.g. considered as an exacerbation of not. 9. In statin therapy, compliance can be measured by seeing if there was a reduction in LDL level before and after treatment initiation. The authors should discuss if this was done or not and if not, why not. 10. Many COPD exacerbations are treated in Accident and Emergency departments without the patient being admitted as an inpatient. Did the authors look at this and if not why not. The relevance of this should be discussed in relation to their study results. 11. It is not unusual for patients to visit more than one Doctor. Did the authors consider that a doctor other than the primary GP may have prescribed statin therapy, please discuss the effect that this would have on the current study results. 12. It is the understanding of this reviewer that since 2004 in the UK, statin medication has been available as 'over-the-counter' medication. The authors should discuss the relevance of this to their current study. 13. In this study the authors mention a 'healthy user' effect, siting flu vaccination. There is no evidence for a healthy user bias as the vaccinations rates are not much different. In a COPD cohort it would be expected that these patients have more doctor visits and as such have more opportunity for vaccinations being given. 14. The authors considered their cohort to be more generalisable than those selected for clinical trials. If so, why include those with no
--	--

	exacerbation in the last month? The authors should exclude these patients and then discuss what the subsequent result is. 15. In Table 3 (page 13), mortality reduced by 50% in a group that should have higher mortality. This finding should be discussed. 16. The finding of 37% lower risk of death was found in a group with more co-morbid disease. Did the authors adjust for all-causes, as it appears to this reviewer that they only adjusted for cardiovascular disease? 17. Table 2 and Table 3. The authors should include the 'N' value and percentages and not only the Hazard Ratio and 95% Confidence Interval. 18. The authors should be more specific about how they consider that the methodology used reduced biases seen in previous studies looking at statin use and exacerbations of COPD (see above comment relating to first vs all exacerbations). The authors should discuss the biases in their own study with relevance to the conclusions they have drawn. 19. The authors should review the RCT article by Schenk P, and colleagues, (Schenk P, Spiel AO, Huettinger F, et al. Can Simvastatin reduce COPD Exacerbations? A Randomised Double Blind Controlled Study. Eur Respir J 2020). that has results that differ from the Criner GJ, etal, STATCPE trial quoted by the authors. the authors should discuss this with reference to their study. 20. In a study of COPD, it would be best to remove the people with missing FEV1 % predicted and incomplete smoking history. It is a mistake to assume someone is a non-smoker (page 8). It is also unwise to impute missing spirometry data. 21. Careful review for typographic errors would be recommended.
--	--

REVIEWER	Dey, Tanujit Brigham and Women's Hospital
REVIEW RETURNED	10-Jun-2021

GENERAL COMMENTS	Overall comment: I appreciate the opportunity to review this manuscript. The goal of the study is novel. However, the current version of the paper needs a substantial improvement in several places as outlined below: Statistical analysis section: Please explain whether assumptions on using the Cox model were verified and reported properly. "People with missing smoking data were assumed non-smokers." -- this assumption needs to be verified by doing a sensitivity analysis: complete data vs imputed data analysis. This is a "missing not at random" situation; proper statistical explanation needed to justify the author's assumption. Sensitivity analysis is required for justifying the use of 90 days for censoring.
--

	Results and Discussion: I don't see a proper reporting of percentage of missing in the variables -- please report them accordingly. Table 1 depicts that the missingness in a few variables are quite high; I am wondering if multiple imputation is good enough in those variables. Please explain this issue either via sensitivity analysis or report results based on complete data only. Please check whether the report related to Table 3 is meant for the secondary analysis; right now it is referring to Table 2.
--	--

VERSION 1 – AUTHOR RESPONSE

Reviewer: 1

Dr. Lies Lahousse, Ghent University

Comments to the Author:

This large, impressive study investigated the association between statins and COPD exacerbations in a general population of patients attending routine primary care.

Thank you

Major comments:

-In the methods, please specify the definition of "research quality data in the current practice" and "exacerbation-specific antibiotics".

We have expanded on the definition of “research quality data” provided in the previous draft, and provided a reference. Exacerbation specific antibiotics are ones that are normally prescribed by Primary care physicians for COPD exacerbations. We used a previously validated algorithm to find exacerbations. We used records of prescriptions for these antibiotics in conjunction with records of exacerbation symptoms or steroid prescription on the same date to identify exacerbations.

-Was the combination of antibiotics and oral steroids on the same date required to meet the definition of exacerbations? I understand that antibiotics alone may complicate the negative control of UTI but it could have underestimated true exacerbation rate?

Yes, the combination of both is intended to identify an exacerbation. This algorithm for identifying exacerbations is described in detail in the cited reference. Negative control means that we are not expecting an association between statin use and UTI. If we were to find one then it might raise concerns about bias somewhere in the study design. We have made this clearer in the description of the study outcomes.

Since in elderly COPD, deterioration due to UTI or due to exacerbation may be hard to distinguish, it is very interesting that UTI was taken as a negative control. However, I wondered how events indicating exacerbations AND UTI within 14 days of each other were classified? Moreover, were UTI and exacerbations analysed as competing risks in time-dependent analyses? Interestingly, adding CVD to the models has an opposite effect for UTI compared to exacerbations.

The primary outcome is exacerbation. A secondary outcome is UTI. Deterioration as such is not an outcome. We do not believe that competing risk analysis is relevant. We did notice the opposite effect on UTI and exacerbations, but as it is not directly relevant to the objectives of this study, we did not bring it up in the Discussion.

Why was FEV1 included as confounder in the association of statins and exacerbations? Was this an attempt to correct for COPD severity which was on average slightly less severe in new statin users? Chronic low-grade system inflammation in COPD is not well associated with FEV1. This could have influenced results if not completely controlled for.

Yes. FEV1 was included as a confounder, as one way of describing COPD severity. In fact, the number of previous exacerbations in the year before the study index date is the COPD-related variable most likely to be associated with subsequent exacerbation rate. Therefore adjusting for the number of previous exacerbations would probably have had the biggest effect on the association between statins and subsequent exacerbation. We adjusted for a wide range of potential confounders; however as this is an observational study, we cannot be certain that the effects of confounding have been fully dealt with.

-How were exacerbation and UTI results when restricted to COPD patients with indications for statin treatment or previous CVD?

As these are data from Primary Care, and statins are mostly prescribed in the Primary Care setting, we would expect that people using statins would have indications for statin treatment. However, those not taking statins would mostly not have such indications. This is why we adjusted for CVD and related variables.

-Follow-up was limited to 3 years because adherence to statins was indeed expected to drop-off over time. However, previous observational studies are suggestive towards beneficial (pleiotropic) effects in COPD after long-term usage? I assume that most statins were initiated between the index date and end of follow-up (confirmed by 1.6y mean time on statins), so the period of actual statin intake may have been too short?

We agree that our conclusions only apply to follow-up of a maximum of 3-years and we have inserted the follow-up time into the discussion. We are not aware of studies of the statin-COPD exacerbation association that have found a stronger association with long-term usage of statins.

How was the time between index date and statin initiation allocated?

Time was allocated to the non-user statin category. We have tried to make this clearer in the statistical analysis section.

-Moreover regarding adherence, since statin-users are more likely to be healthy / more adherent users (also reflected by their higher influenza vaccinations and visiting), they are also likely to be more adherent to COPD maintenance therapy and therefore be less exacerbation-prone (confirmed by having 2% less exacerbations the year before index)?

We did note in the Results that COPD appeared slightly less severe in statin users compared to non-users. We adjusted for number of previous exacerbations and other variables related to COPD severity in some of the analyses which should have dealt with confounding by COPD severity. As this is an observational study we cannot be certain that we have completely controlled for such effects.

Minor comments:

- More likely to be current smokers (8% vs. 10%)?

Thank you. Corrected.

- Please specify in results text which factors are included in the "variables related to severity of COPD"

This has already been done in the original manuscript. See Covariates section of Methods. It is repeated in the legend to Table 2.

- Lowering mortality and severe exacerbation requiring hospitalisation are very important outcomes in COPD, which seem currently underestimated in the overall conclusions.

Our findings were similar for exacerbation whether severe or not. Death was included as a secondary outcome so that we could compare results with other studies. A substantial difference might indicate some source of bias in our study.

Reviewer: 2

Dr. Raheleh Roudi, Iran University of Medical Sciences, Iran University of Medical Sciences

Comments to the Author:

In the current study, Smith et al evaluated the statin prescription in patients with chronic obstructive pulmonary disease and risk of exacerbations. The authors need to consider some potential points to improve the current version;

1) COPD could be a driving factor in lung cancer and the authors should add this point as an important aspect of this study due to unefficient treatment options in lung cancer using the related articles such as Jianwei Zhu, et al - 2017 - Alfredo Tartarone, et al - 2019 - Monireh Mohsenzadegan, et al - 2020 and Fausto Petrelli, et al - 2021.

We could not find relevant papers from the information above, but also we could not see the relevance of any relationship between COPD and lung cancer and treatment for lung cancer.

2) The association of statins with other diseases such as lung cancer have been suggested. The authors need to add these related data such as Feng Li, et al - 2018 and so on.

We could not find the paper by Feng Li. We have now cited two other papers in the Discussion that consider the broad range of other outcomes that might be affected by taking statins.

Reviewer: 3

Dr. Raewyn Hopkins, Auckland University of Technology

Comments to the Author:

General Comment:

The researcher's present a retrospective cohort study comparing newly prescribed statin users, with non-statin users, following the selection of an index date for study start. The researchers used a selection of General Practices in England who contribute data to the Clinical Practice Research Datalink, which is linked to a hospital inpatient statistics. The observation period was only an average of 2.5 years.

We did an intention to treat analysis (patients were assumed to continue as statin users after their first prescription of statins) with follow-up limited to 3-years. We did this because we thought that adherence to statins would drop off over time. We have added a sentence in the statistical analysis to explain this.

Specific comments:

1. Included in the cohort are patients with Asthma, other respiratory disease and lung cancer, all of which effect lung function results and the development of chest infections / exacerbations. These patients should be removed from the analysis.

Our aim was to do this study for a real-world population so we did not want to remove these people from the study population. We adjusted for various comorbidities, including these conditions. This should control for confounding if having these comorbidities is associated with taking statins, and the subsequent outcome.

2. Given that smoking is a leading cause of COPD and cardiovascular disease, patients with missing smoking history should be removed as to assume them to be non-smokers maybe be incorrect.

Complete case analysis can also cause a biased estimate. Please also see response to Reviewer 3 point 20.

3. In the introduction (paragraph 1, page 5), the authors mention inflammatory biomarkers in relation to COPD. It is not clear to this reviewer whether the authors looked inflammatory markers such as CRP levels before and after statin treatment. The authors should do this and discuss the results.

No, we did not do this. One of the disadvantages of using electronic health records is that test results are not usually available at required times for all study participants. Such a study is unlikely to be straightforward. In addition, the exacerbation outcome is directly relevant to patients.

4. There are studies showing evidence that concurrent smoking may delay the anti-inflammatory effect of statins, especially in the airways. The authors should stratify the groups by current versus ex-smokers and report and discuss the results.

Given the overall null result for the association of exacerbation with statin use we did not pursue subgroup analyses. It is likely that any statistically significant results would be due to multiple testing, and affected by low power. Note only about 10% of the population were current smokers.

5. Table 1 (page 9-11), shows the exacerbations within the year before the index date. The authors should stratify the cohorts according to exacerbations the year before the index date and see if the lack of a statin association is in those with 1 or more exacerbation.

See response to point 4.

6. In a cohort study such as this, the authors should add in and analyse all exacerbations. Time to 1st exacerbations is not useful unless it is an RCT. The authors should explain and discuss why they dropped out moderate exacerbations from analysis.

We looked at any exacerbation, and severe exacerbations, so we did not “drop moderate exacerbations”. The any exacerbation outcome was analysed in two ways: as a standard Cox regression censoring at the first event, and with an Andersen Gill model that allowed all exacerbations to be included in the analysis model. We did not find any substantive difference between this analysis and the Cox regression with the first exacerbation as the outcome. Cox proportional hazards analysis using time-to-event is a very common statistical method for assessing outcomes in epidemiological studies using large datasets.

7. The authors should discuss how a diagnosis of pneumonia was handled, that is was it considered to be an exacerbation in this study.

We did not record diagnosis of pneumonia in this study. However, there will be overlap in definitions of exacerbations and pneumonia because these have similar clinical presentations and management. The exacerbation outcome (which we based on the validated definition of exacerbations) does not include coded diagnoses of pneumonia in Primary Care, but would include a symptom code plus antibiotic prescription (e.g. ‘cough’ plus amoxicillin prescription could be either exacerbation or pneumonia).

Hospitalised exacerbations were identified from the HES inpatient record using acute COPD or an acute respiratory code as the primary diagnosis of hospitalisation, as hospital coding is imperfect and this was thought on the balance to be more sensitive than including exacerbation codes only. We have added the ICD10 codes that were used to indicate a severe COPD exacerbation to the text of the manuscript.

8. Given the cardiac profile of the cohorts, the authors should discuss how a diagnosis of heart failure that received an antibiotic was handled, e.g. considered as an exacerbation or not.

We used antibiotic prescriptions in conjunction with other indicators of an exacerbation (prescription of steroid or exacerbation symptoms). Under this definition, we cannot be totally certain that we are correctly identifying all exacerbations and not classifying any other events as exacerbations. However, the algorithm that we used has been validated previously. In UK primary care, a patient with COPD receiving an exacerbation-specific antibiotic is much more likely to be receiving it for a COPD exacerbation than any other reason. However such assumptions are a necessary limitation of using electronic health data. Please also see the response to referee 1 about exacerbation specific antibiotics.

9. In statin therapy, compliance can be measured by seeing if there was a reduction in LDL level before and after treatment initiation. The authors should discuss if this was done or not and if not, why not.

This will be hard to do for electronic health record data as such test results are not likely to be recorded at the required intervals.

10. Many COPD exacerbations are treated in Accident and Emergency departments without the patient being admitted as an inpatient. Did the authors look at this and if not why not. The relevance of this should be discussed in relation to their study results.

We have already mentioned this in the Discussion. Our PPI contributor suggested that we look for COPD exacerbations in A & E data, but when we investigated how events are recorded in A & E data, we found that A & E records do not contain sufficient detail to identify COPD exacerbations. Documentation is sent to a patient's GP after an A&E department or out-of-hours GP visit, and if this states an exacerbation of COPD it is likely that this would be coded on to the primary care records, although this would vary between practices.

11. It is not unusual for patients to visit more than one Doctor. Did the authors consider that a doctor other than the primary GP may have prescribed statin therapy, please discuss the effect that this would have on the current study results.

In the UK, patients are usually registered at a single GP practice. The data that we used includes all contacts with GPs, so it does include data relating to consultations with, and drugs prescribed by any GP within the patients' practice. It would be extremely unusual for a long-term medication such as a statin to be initiated or prescribed anywhere else other than by the GP.

12. It is the understanding of this reviewer that since 2004 in the UK, statin medication has been available as 'over-the-counter' medication. The authors should discuss the relevance of this to their current study.

If those not receiving statin prescriptions in Primary Care, were instead taking over-the-counter statin medication (OTC), this could potentially reduce the observed effect size. However, OTC statin medication is limited to 10mg simvastatin. Most patients in our study were prescribed 40mg simvastatin. In addition, the support for OTC statins by pharmacists and clinicians has been limited, and the take-up of OTC statins has been low. Therefore, the impact on our results is probably very small. We have added a comment on this to study limitations.

13. In this study the authors mention a 'healthy user' effect, citing flu vaccination. There is no evidence for a healthy user bias as the vaccination rates are not much different. In a COPD cohort it would be expected that these patients have more doctor visits and as such have more opportunity for vaccinations being given.

We discuss a healthy user effect and include influenza vaccination as a covariate. But we did not link the two in the manner suggested by the referee. We do not think it is necessary to speculate about reasons for differences in baseline variables between the exposure groups.

14. The authors considered their cohort to be more generalisable than those selected for clinical trials. If so, why include those with no exacerbation in the last month? The authors should exclude these patients and then discuss what the subsequent result is.

By generalizable we mean that our cohort is more typical of the UK population with COPD. In other words, we have not applied such strict inclusion criteria to the study population as many trials do. The population of those who HAVE had an exacerbation in the last month would be very small indeed and highly atypical of primary care COPD, given that half of COPD patients had no exacerbation in the year prior to index date (Table 1). Please also see the answer to point 5, regarding subgroup analyses.

15. In Table 3 (page 13), mortality reduced by 50% in a group that should have higher mortality. This finding should be discussed.

We discussed the findings relating to the outcome of death and adjusting for time-dependent CVD covariates in relation to Table 2.

16. The finding of 37% lower risk of death was found in a group with more co-morbid disease. Did the authors adjust for all-causes, as it appears to this reviewer that they only adjusted for cardiovascular disease?

The 37% lower risk of death was found for both adjustments.

17. Table 2 and Table 3. The authors should include the 'N' value and percentages and not only the Hazard Ratio and 95% Confidence Interval.

We have added the number of events for nonusers and users of statins to Table 2. We have added the total follow up time before censoring or death by statin use to the text.

18. The authors should be more specific about how they consider that the methodology used reduced biases seen in previous studies looking at statin use and exacerbations of COPD (see above comment relating to first vs all exacerbations). The authors should discuss the biases in their own study with relevance to the conclusions they have drawn.

Please see the response to point 6 regarding first/all exacerbations. We think that we have described the methodology that we have used to reduce bias. We restricted analysis of statins to new users to remove prevalent user bias, we adjusted for a wide range of potential confounders, we included control outcomes, and we used a time-varying exposure to attribute time to the correct exposure status and therefore to avoid immortal time bias. Also, using electronic healthcare records from Primary Care means that the population in the study and their use of statins is reflective of statin use by people with COPD in a real-world setting. Please also see the section on strengths and limitations in the Discussion where we consider the limitations in relation to the conclusions drawn.

19. The authors should review the RCT article by Schenk P, and colleagues, (Schenk P, Spiel AO, Huettinger F, et al. Can Simvastatin reduce COPD Exacerbations? A Randomised Double Blind Controlled Study. Eur Respir J 2020). that has results that differ from the Criner GJ, et al, STATCPE trial quoted by the authors. the authors should discuss this with reference to their study.

Thank you for alerting us to this study. We have included it in the Discussion.

20. In a study of COPD, it would be best to remove the people with missing FEV1 % predicted and incomplete smoking history. It is a mistake to assume someone is a non-smoker (page 8). It is also unwise to impute missing spirometry data.

Complete case analysis is also not necessarily correct. It reduces power and can result in biased results. Also analyses with different adjustments contain different people and are therefore difficult to compare (unless we do all analyses on the smallest data set). For smoking we hypothesise that this is a missing not a random situation (GPs are less likely to ask about smoking in people they think do not smoke) and therefore standard multiple imputation methods would not be applicable. This is why we designated missing values to the non-smoking category. There were only about 200 missing values of smoking status in the cohort of 48,000. We agree that FEV1 could potentially be missing not at random too. However, we did also repeat main analyses as complete case analyses, and found almost no difference in results. We have now stated in the results of the revised paper that we repeated analyses as complete case analyses. Based on this reviewer comment we have also repeated some main analyses but excluding the FEV1 covariate. Again, we found almost no difference in results from those presented in Table 2.

21. Careful review for typographic errors would be recommended.

Reviewer: 4

Dr. Tanujit Dey, Brigham and Women's Hospital

Comments to the Author:

Overall comment: I appreciate the opportunity to review this manuscript. The goal of the study is novel. However, the current version of the paper needs a substantial improvement in several places as outlined below:

Statistical analysis section:

Please explain whether assumptions on using the Cox model were verified and reported properly.

For the time-independent covariates the proportional hazards assumption was assessed graphically using -ln survival curves for each covariate. We have added this statement to the text in the revised version. By definition the PH assumption is not satisfied for statin use as it is a time-dependent variable.

"People with missing smoking data were assumed non-smokers." -- this assumption needs to be verified by doing a sensitivity analysis: complete data vs imputed data analysis. This is a "missing not at random" situation; proper statistical explanation needed to justify the author's assumption.

Please see responses to Reviewer 3 points 2 and 20.

Sensitivity analysis is required for justifying the use of 90 days for censoring.

The censoring at 90-days is in itself a secondary analysis or sensitivity analysis. We thought that statin adherence might drop with time since starting to use statins. Therefore investigating the association over a shorter follow-up might yield a stronger association. We do not feel that further sensitivity analyses of different time points for censoring is justified.

Results and Discussion:

I don't see a proper reporting of percentage of missing in the variables -- please report them accordingly. Table 1 depicts that the missingness in a few variables are quite high; I am wondering if multiple imputation is good enough in those variables. Please explain this issue either via sensitivity analysis or report results based on complete data only.

There are none/few codes in electronic health record for absence of a condition or treatment. Therefore, missing values of comorbidities or treatments were interpreted as meaning no comorbidity or no treatment. We have stated this in the text of the revised manuscript. Please also see response to referee 3, points 2 and 20.

Please check whether the report related to Table 3 is meant for the secondary analysis; right now it is referring to Table 2.

What we have done looks to be correct. We have divided Table 2 into Primary and Secondary outcomes in the revised manuscript, which we hope will make the table clearer.

VERSION 2 – REVIEW

REVIEWER	Lahousse, Lies Ghent University
REVIEW RETURNED	03-Sep-2021

GENERAL COMMENTS	Thanks to the authors for revising and improving their manuscript! Because a smoking stratified and competing risk analysis might have added to the paper, I would suggest to mention those in the discussion/limitations, together with the lack of inflammatory biomarkers such as CRP and that the true moderate exacerbation rate might have been underestimated since prescriptions of antibiotics without oral steroids were not counted as exacerbation.
---

REVIEWER	Roudi, Raheleh University of Minnesota Twin Cities
REVIEW RETURNED	12-Aug-2021

GENERAL COMMENTS	The previous comments have not been considered.
---

REVIEWER	Hopkins, Raewyn Auckland University of Technology
REVIEW RETURNED	25-Aug-2021

GENERAL COMMENTS	The authors have responded to all reviewers comments. The manuscript has been improved to reflect the reviewers comments.
--

VERSION 2 – AUTHOR RESPONSE

Reviewer: 2

Dr. Raheleh Roudi, University of Minnesota Twin Cities

Comments to the Author:

The previous comments have not been considered.

Please see our previous response which we believe addressed the reviewer comments.

Reviewer: 3

Dr. Raewyn Hopkins, Auckland University of Technology

Comments to the Author:

The authors have responded to all reviewers comments.

The manuscript has been improved to reflect the reviewers comments.

Thankyou

Reviewer: 1

Dr. Lies Lahousse, Ghent University

Comments to the Author:

Thanks to the authors for revising and improving their manuscript!

Because a smoking stratified and competing risk analysis might have added to the paper, I would suggest to mention those in the discussion/limitations, together with the lack of inflammatory biomarkers such as CRP and that the true moderate exacerbation rate might have been underestimated since prescriptions of antibiotics without oral steroids were not counted as exacerbation.

Thank you for these comments

The objective of this study was to assess the overall association between taking statins and subsequent COPD exacerbations. We did not have a hypothesis that the associations would differ between strata of smoking. Also given the overall null result for the association of exacerbation with statin use we also did not pursue any subgroup analyses. We do not see a need to comment on this the discussion.

In our response to a previous comment by this referee we said that the prevalence of current smoking was 10%. In fact we had labels mixed up in our Table 1, this is the prevalence of never smokers. We apologise for this, and we have corrected the mistake.

Special statistical methods for dealing with competing risk are not needed as we were examining the association between an exposure and outcome. The statistical analysis method that we have used where we treat death as censored is the correct method of analysis for aetiologic studies. There is therefore no need to add any points to the discussion about the use/non use of competing risk methods.

We have already mentioned in the Introduction (p3) that statins may have anti-inflammatory properties. Our study objective was to assess whether using statins affected disease severity in COPD.

Thank you. We have added the following comment about potential misclassification of COPD exacerbations to the limitations section on p15. "*COPD exacerbations are recorded in a number of different ways in Primary Care records and are therefore not straightforward to identify. We used a validated algorithm to identify exacerbations, but there is still likely to have been some misclassification.¹⁷ Non-differential misclassification is expected to move effect sizes towards the null.*"